# [Re] Exacerbating Algorithmic Bias through Fairness Attacks

## Reproducibility Summary

**Scope of Reproducibility**

We conducted a reproducibility study of the paper *Exacerbating Algorithmic Bias through Fairness Attacks* [11]. According to the paper, current research on adversarial attacks is primarily focused on targeting model performance, which motivates the need for adversarial attacks on fairness. To that end, the authors propose two novel data poisoning adversarial attacks, the influence attack on fairness and the anchoring attack. We aim to verify the main claims of the paper, namely that: a) the proposed methods indeed affect a model's fairness and outperform existing attacks, b) the anchoring attack hardly affects performance, while impacting fairness, and c) the influence attack on fairness provides a controllable trade-off between performance and fairness degradation.

**Methodology**

We chose PyTorch Lightning to re-implement all of the code required to reproduce the original paper's results. Our implementation enables the quick and easy extension of existing experiments, as well as the integration with the various development tools that come with PyTorch Lightning. All of our experiments took about 120 hours to complete on a machine equipped with an Intel Core i7 7700k CPU and an NVIDIA GeForce GTX 1080 GPU.

**Results**

Our results slightly deviate from the ones reported by the authors. This could be attributed to the design choices we had to make, due to ambiguities present in the original paper. After inspecting the provided codebase along with relevant literature, we were able to replicate the experimental setup. In our experiments, we observe similar trends and hence we can verify most of the paper's claims, albeit not getting identical experimental results.

**What was easy**

The original paper is well-structured and easy to follow, with the principle ideas behind the proposed algorithms being very intuitive. Additionally, the datasets used in the experiments are publicly available, small in size, and the authors provide their code on GitHub.

**What was difficult**

During our study, we encountered a few unforeseen issues. Most importantly, we were not able to identify critical technical information required for the implementation of the proposed algorithms, as well as a detailed description of the models used, their training pipeline, hyperparameters, and data pre-processing techniques. Furthermore, the publicly available code is convoluted and employs out-of-date libraries, making it difficult to set up the necessary environment.

**Communication with original authors**

We contacted the paper's first author once to confirm our understanding of certain elements of the paper that were either not specific enough or missing. Although they responded fairly quickly, their answer prompted us back to the paper and the provided codebase, while not encouraging any further communication.

Submitted to ML Reproducibility Challenge 2021. Do not distribute.

# 1 Introduction

Adversarial attacks have become popular in the machine learning community since they allow scientists to understand and mitigate the weaknesses of the employed models. Current research is primarily focused on adversarial attacks targeting the performance of machine learning systems [3, 10], but recent studies indicate that adversarial attacks can also be used to target fairness [11, 12, 13]. In the studied paper, the authors propose two novel families of adversarial attacks - the influence attack on fairness and the anchoring attack - and demonstrate their effect in exacerbating algorithmic bias by evaluating them on three datasets using two well-known fairness metrics.

Both of the proposed methods belong to the family of data poising attacks, in which the adversary attempts to inject malicious data points into the training data. In particular, given a "clean" training dataset $\mathcal{D}_c$, i.e. a dataset containing only the original training samples, the adversary generates a "poisoned" dataset $\mathcal{D}_p$ and integrates it into the original one, resulting in the final train set $\mathcal{D}_{\text{train}} = \mathcal{D}_c \cup \mathcal{D}_p$. The poisoned dataset $\mathcal{D}_p$ is generated in such a way that training with $\mathcal{D}_{\text{train}}$ results in a model with degraded performance or, in our case, a less fair model.

The paper considers a binary classification scenario, under a common fairness setup with two demographic groups; the advantaged $\mathcal{D}_{\text{adv}}$ and the disadvantaged $\mathcal{D}_{\text{disadv}}$. Under this setting and given an adversarial loss that increases when the model makes unfair decisions, the influence attack on fairness finds adversarial data points by performing gradient ascent on the adversarial loss. On the other hand, the anchoring attack places poisoned points in the close vicinity of two target points, one from $\mathcal{D}_{\text{adv}}$ and one from $\mathcal{D}_{\text{disadv}}$, with the opposite labels but the same demographic.

# 2 Scope of reproducibility

In this reproducibility study we aim to verify the following main claims of the paper:

- Both of the proposed attacks impact the fairness of the targeted model, outperforming other attacks in the literature, such as Koh's basic influence attack [9] and Solan's gradient-based poisoning attack [13].

- The anchoring attack has little to no impact on the model's accuracy, making it more difficult to detect.

- The influence attack on fairness provides a controllable trade-off between the impact on performance and fairness via a regularization term $\lambda$.

Additionally, we extend the evaluation set up to test whether current methods can be used to invert the inherent bias of a dataset. To this end, we re-implement the entire experimental setup, and hence contribute:

- an extensive study and evaluation of the adversarial attacks proposed by Mehrabi et al. [11].

- a modification to the influence attack on fairness which can invert or diminish the inherent bias of a dataset.

- a comprehensible and easily extensible codebase, which can be used both in the evaluation of current methods and as a framework for further research on adversarial attacks on fairness.

# 3 Methodology

## 3.1 Poisoning Attacks

Poising attacks are a category of adversarial attacks where the attacker impacts a system by injecting a small portion of engineered malicious data into its training set. In particular, we consider that the system is trained on a clean dataset $\mathcal{D}_c$ and evaluated on a test dataset $\mathcal{D}_{\text{test}}$. The attacker has knowledge of both sets, as well as of the system's architecture and its training pipeline. With this information, the attacker creates a poisoned dataset $\mathcal{D}_p$, with $|\mathcal{D}_p| = \epsilon |\mathcal{D}_c|$, so that training the attacked system on $\mathcal{D}_c \cup \mathcal{D}_p$ impacts its performance, or in our case its fairness. The parameter $\epsilon$ controls the percentage of poisoned points, which depends on the nature of the application. Finally, we assume that the attacked system has a defense mechanism $B$ that possibly removes poisoned data with the use of anomaly detection techniques.

### 3.1.1 Influence Attack on Fairness

The Influence Attack on Fairness (IAF) is a gradient-based data poisoning attack, derived from a combination of the works of Koh et al. [8], which introduces the basic influence attack, and Zafar et al. [15], which proposes a novel

fairness loss. The main idea is to build $\mathcal{D}_p$ from copies of two datapoints $(\tilde{\mathbf{x}}_1, \tilde{y}_1)$ and $(\tilde{\mathbf{x}}_2, \tilde{y}_2)$ sampled from $\mathcal{D}_c$, and progressively update them to decrease model fairness, as measured by an adversarial loss $\mathcal{L}_{\mathrm{adv}}$. The authors propose to use $\mathcal{L}_{\mathrm{adv}} = \mathcal{L}_{bc} + \lambda \cdot \mathcal{L}_f$, where $\mathcal{L}_{bc}$ is any binary classification loss and $\mathcal{L}_f$ is the aforementioned fairness loss.

To update $(\tilde{\mathbf{x}}_1, \tilde{y}_1)$ and $(\tilde{\mathbf{x}}_2, \tilde{y}_2)$, the paper suggests to perform gradient ascent on $\mathcal{L}_{\mathrm{adv}}$ and then update $\mathcal{D}_p$ with their copies. Since $\mathcal{L}_{\mathrm{adv}}$ depends on the trained model's parameters $\hat{\boldsymbol{\theta}}$, the gradient ascent follows an expectation-maximization scheme, where in the expectation step the model is trained on $B(\mathcal{D}_c \cup \mathcal{D}_p)^1$and in the maximization step the points move on the gradient direction. Although this idea is very intuitive, calculating the gradient of $\mathcal{L}_{\mathrm{adv}}$ w.r.t each adversarial point is challenging. The approach presented in [9] is to apply the chain rule as $\frac{\partial \mathcal{L}}{\partial \tilde{\mathbf{x}}_i} = \frac{\partial \mathcal{L}}{\partial \hat{\boldsymbol{\theta}}} \frac{\partial \hat{\boldsymbol{\theta}}}{\partial \tilde{\mathbf{x}}_i}$, with the later derivatives calculated in Equations 1 and 2. Here, $\ell$ is the model's train loss for the single data point and $H_{\hat{\boldsymbol{\theta}}}$ is the Hessian of the train loss at $\hat{\boldsymbol{\theta}}$ w.r.t. the adversarial sample $\tilde{\mathbf{x}}_i$. More details for the derivation of these formulas, as well as how to compute them efficiently, can be found in Section 2.2 of [8] and Section 4.1.1 of [9].

$$g_{\hat{\boldsymbol{\theta}}, \mathcal{D}_{\mathrm{test}}} \stackrel{\mathrm{def}}{=} \frac{\partial \mathcal{L}}{\partial \hat{\boldsymbol{\theta}}} = \frac{1}{|\mathcal{D}_{\mathrm{test}}|} \sum_{(\mathbf{x}, y) \in \mathcal{D}_{\mathrm{test}}} \nabla \ell(\hat{\boldsymbol{\theta}}; \mathbf{x}, y) \quad (1) \qquad \frac{\partial \hat{\boldsymbol{\theta}}}{\partial \tilde{\mathbf{x}}} = -H_{\hat{\boldsymbol{\theta}}}^{-1} \frac{\partial^2 \ell(\hat{\boldsymbol{\theta}}; \tilde{\mathbf{x}}, \tilde{y})}{\partial \hat{\boldsymbol{\theta}} \partial \tilde{\mathbf{x}}} \qquad (2)$$

### 3.1.2 Anchoring Attack

The anchoring attack places poisoned datapoints, which act as anchors, in the near vicinity of two target points. In particular, the attacker samples two target points $\mathbf{x}_{\mathrm{target}-}$, and $\mathbf{x}_{\mathrm{target}+}$ from the advantaged $\mathcal{D}_{\mathrm{adv}}$ and disadvantaged $\mathcal{D}_{\mathrm{disadv}}$ groups of the train dataset. Subsequently, $|\epsilon n|$ poisoned datapoints $\{\tilde{\mathbf{x}}_i\}_{i=1}^{|\epsilon n|}$ are generated in the near vicinity of the target points, placing them in the same demographic group but on opposite categories $\tilde{y}_i \neq y_{\mathrm{target}}$. Intuitively, this aims to move the decision boundary so that more advantaged points have a positive predictive outcome and more disadvantaged points have a negative outcome, hence inducing more biased outcomes.

The paper proposes two methods to sample $\mathbf{x}_{\mathrm{target}-}$ and $\mathbf{x}_{\mathrm{target}+}$ from the dataset:

- **Random Anchoring (RAA)**: $\mathbf{x}_{\mathrm{target}}$ is sampled uniformly for each demographic group.

- **Non-Random Anchoring (NRAA)**: $\mathbf{x}_{\mathrm{target}}$ is the point close to the most similar points given its label and demographic. This aims to affect as many points as possible when placing poisoned points within its vicinity.

In the latter case, the authors suggest to consider $\mathbf{x}$ and $\mathbf{x}'$ neighbors if and only if $||\mathbf{x} - \mathbf{x}'|| < R,\ R \in \mathbb{R}$. The choice of $R$ and the specific norm $|| \cdot ||$ is not defined in the paper. After careful examination of the provided code, we found that the L1 norm was used and the $R$ values were hard-coded for each dataset. To avoid manual experimentation for each dataset's $R$, we propose the following definition for the most popular point in a dataset $\mathcal{X}$:

$$\mathbf{x}_{\mathrm{pop}} \stackrel{\mathrm{def}}{=} \underset{\mathbf{x} \in \mathcal{X}}{\mathrm{argmax}} \sum_{\mathbf{x}' \in \mathcal{X}} \exp\left( -\frac{d(\mathbf{x}, \mathbf{x}')}{\sigma_{d(\mathcal{X})}^2} \right) \qquad (3)$$

where $d$ is a distance metric and $\sigma_{d(\mathcal{X})}^2$ denotes the variance of the points' distances to each other under $d$. Motivation for this choice and implementation details can be found in Appendix B.

### 3.2 Defenses

The authors use a defense mechanism $B$ in both of the proposed attacks, along with a corresponding projection function that bypasses it, without specifying the actual type of the defense. Although this information is not crucial for the comprehension of the attacks, we deem it critical for their reproducibility.

After inspecting the code and the cited literature, we found that the defense mechanism used is a combination of the L2 defense and the slab defense [14]. The L2 defense removes points far from their corresponding class' centroid according to the $L_2$ distance: $\boldsymbol{\beta}_y = \mathbb{E}_{\mathcal{D}}[\mathbf{x} \mid y],\ s_{\boldsymbol{\beta}} = ||\mathbf{x} - \boldsymbol{\beta}_y||_2$. The slab defense projects points onto the line between the class centroids and then removes the points too far from the centroids: $\boldsymbol{\beta}_y = \mathbb{E}_{\mathcal{D}}[\mathbf{x} \mid y],\ \mathbf{s}_{\boldsymbol{\beta}} = \left|(\boldsymbol{\beta}_1 - \boldsymbol{\beta}_{-1})^\top (\mathbf{x} - \boldsymbol{\beta}_y)\right|$.

---

[1]In the original paper, the authors mention that training is performed on $\mathcal{D}_c \cup \mathcal{D}_p$, but we deem that using $B(\mathcal{D}_c \cup \mathcal{D}_p)$ is more sensible and congruent with the basic influence attack [9].

The feasible set $\mathcal{F}_\beta \subset \mathcal{X} \times \mathcal{Y}$ encodes the defenses, as well as the constraints for the input's features, and contains all of the points that would not be discarded by the defender. For the L2 constraint, we apply the LP relaxation technique as described in [9] and end up with a feasible set $\mathcal{F}_{LP} = \left\{ (\mathbf{x}, y) : \mathbb{E}\left[ \left\| \hat{\mathbf{x}} - \boldsymbol{\mu}_y \right\|_2^2 \right] \leq \tau_y^2 \wedge \mathbf{x} \in \mathbb{R}_{\geq 0} \right\}$, where $\boldsymbol{\mu}_y$ denotes the centroid of the subset of points in class $y$. The parameter $\tau_y$ is chosen dynamically for each $y$, such that 90% of the points in the $\mathcal{D}_y$ subset satisfy the L2 constraint. For the slab constraint, we construct a feasible set $\mathcal{F}_{\text{slab}} = \left\{ (\mathbf{x}, y) : |(\boldsymbol{\mu}_1 - \boldsymbol{\mu}_{-1})^\top (\mathbf{x} - \boldsymbol{\mu}_y)| \leq \tau_y' \wedge x \in \mathbf{R}_{\geq 0} \right\}$, where $\boldsymbol{\mu}_1$ and $\boldsymbol{\mu}_{-1}$ denote the centroids of classes 1 and $-1$ respectively. Once again, the parameter $\tau_y'$ is chosen dynamically for each $y$ such that 90% of the points in the $\mathcal{D}_y$ subset satisfy the slab constraint. Our final feasible set is the intersection of the feasible sets under the two constraints, plus any additional input constraints imposed by $\mathcal{X}$.

Projecting points onto $\mathcal{F}_\beta$ takes the form of an optimization problem, namely $\text{argmin}_{\mathbf{x} \in \mathcal{F}_\beta} \|\mathbf{x} - \tilde{\mathbf{x}}_i\|_2$, where $\tilde{\mathbf{x}}_i$ denotes the poisoned point. We then simply solve the optimization problem using the library CVXPY with the SCS solver. This procedure is extensively discussed in [9], Section 3.3.

# 4 Experimental Setup

## 4.1 Model and training pipeline

We did not manage to find a detailed description of either the model used or its training pipeline in the original paper. The authors mention that the hinge loss was used, leading us to assume that benchmarked model was a Support Vector Machine. However, after examining their code, we identified that the default model used was a Logistic Regression model. We also followed this choice, as it allows for an easy calculation of the fairness loss used in the influence attack on fairness. Additionally, the authors seem to use SciPy's `fmin_ncg` optimizer to train the model, which is a second-order optimization algorithm that uses conjugate gradients. In our implementation, we opted for Stochastic Gradient Descent, which should be able to converge to the same parameters, as the minimization problem is convex. In our reported results, we used the average over three runs to account for any stochasticity in the pipeline.

## 4.2 Datasets

We carry out our experiments on the same three datasets as the original paper and consider "gender" to be the sensitive attribute. We use a pre-processed version of each dataset, as provided by the authors, to have a common starting point. However, we later discovered a few issues regarding the pre-processing pipeline, which we elaborate on in Appendix A. In all cases, the test set consists of 20% of the total data and there is no validation set. A short description of each dataset is presented below:

**German Credit Dataset**[2] [7]. This dataset has 1000 entries of loan applicants. Each applicant is characterized by 13 categorical and 7 numerical features describing their credit risk and is classified as either "good" or "bad", in terms of their ability to repay the loan.

**COMPAS Dataset**[3] [1]. This dataset has 7214 entries of criminal defendants. We utilize 8 categorical features from the dataset to predict whether a defendant will recommit a crime within 2 years.

**Drug Consumption Dataset**[4] [5]. This dataset has 1885 entries of people alongside their drug history. Each person is described by 13 numerical attributes, which can be used to infer drug usage of 18 different substances. We focused on predicting whether individuals have used cocaine in their lifetime, akin to the original paper.

## 4.3 Fairness Metrics

We evaluate the impact of our attacks both in terms of performance and fairness. For performance, we use the accuracy error, while for fairness we use the Statistical Parity Difference (SPD) [4] and the Equality of Opportunity Difference (EOD) [6]. This evaluation protocol matches the one in the original paper, although our implementation of EOD gives different results. We were able to verify our results' validity by comparing them with the AI Fairness 360 library [2].

---

[2] https://archive.ics.uci.edu/ml/machine-learning-databases/statlog/german/german.data-numeric
[3] https://github.com/propublica/compas-analysis/blob/master/compas-scores-two-years.csv
[4] https://archive.ics.uci.edu/ml/machine-learning-databases/00373/drug_consumption.data

Moreover, the original paper used the absolute values of the aforementioned metrics, which we followed for the reproduced experiments but not for our extensions, as the metrics' signs contained the necessary information.

**Statistical Parity Difference**. Statistical parity is used to ensure that the demographic distribution of the samples being classified positively (or negatively) is similar to the distribution of the entire population. As a result, when we measure the difference in statistical parity between the two demographics (advantaged and disadvantaged groups), we can deduce whether a model is biased in favoring or harming one of the two groups.

$$\text{SPD} = \mid P(y_{\text{pred}} = +1 \mid \mathbf{x} \in \mathcal{D}_{\text{adv}}) - P(y_{\text{pred}} = +1 \mid \mathbf{x} \in \mathcal{D}_{\text{disadv}}) \mid$$

**Equality of Opportunity Difference**. Equality of opportunity is used to guarantee that samples with a positive ground truth label are just as likely to be classified positively, regardless of the demographic group they belong in. By measuring the difference in equality of opportunity for the two groups, we can identify whether the model is biased towards classifying positively more often for either demographic group, given that they have a positive ground-truth label.

$$\text{EOD} = \mid P(y_{\text{pred}} = +1 \mid \mathbf{x} \in \mathcal{D}_{\text{adv}}, y_{\text{label}} = +1) - P(y_{\text{pred}} = +1 \mid \mathbf{x} \in \mathcal{D}_{\text{disadv}}, y_{\text{label}} = +1) \mid$$

### 4.4 Hyperparameters

For all of our experiments, we trained the models for 300 epochs with early stopping based on the train accuracy. We chose an SGD optimizer with a learning rate of 0.001, weight decay of 0.09, and batch sizes of 10, 50, and 10 for the German Credit, Drug Consumption, and COMPAS datasets respectively. Regarding the adversarial attack hyperparameters, we used 100 iterations and a step size $\eta = 0.01$ for the IAF, and $\tau = 0$ for both anchoring attacks.

### 4.5 Implementation Details

We implemented the data poisoning attacks described above in Python, using PyTorch Lightning to train our models[5]. Each attack, along with its helper functions, is implemented in a separate file under the `attacks` folder. We also placed a `utils.py` file under the same folder, which implements essential utilities that are leveraged by all adversarial attacks. We defined two abstract classes, `Dataset` and `Datamodule`, in the corresponding files under the `datamodules` folder, which enable our framework to process a dataset from a given file and construct the required PyTorch `DataLoader` objects. Consequently, each dataset mentioned in Section 4.2 corresponds to a separate file under the same folder, deriving from the `Datamodule` class. Our models are placed under the `models` folder, deriving from the `LinearModel` class, while the training pipeline is described in `trainingmodule.py`. Finally, our fairness metrics and losses are available in `fairness.py`.

In this way, besides providing a well-structured and easy-to-follow code, we also allow fellow researchers to extend our experiments by easily incorporating different models, attacks, datasets, and fairness metrics. To implement a new attack, one can simply create a separate file under the `attack` folder and leverage the implemented attack utilities, such as the projection and defense mechanisms. To test existing attacks with a different dataset, one can create a new PyTorch Lightning `LightningDatamodule` that extends our `Datamodule` class. Finally, in order to test a different model, one needs to create a PyTorch `Module` that extends the `LinearModel` class and update the `BinaryClassifier` class accordingly.

### 4.6 Computational requirements

All of our experiments required a total of 120 hours on a machine with an Intel Core i7 7700k CPU and an NVIDIA GeForce GTX 1080 GPU. We found the most computationally expensive part to be the training of the models, and hence the influence attack, which requires multiple train iterations. This makes it significantly slower than the anchoring attack. However, it is worth noting that a GPU is not strictly necessary. GPU speedups were in the vicinity of 20% over a CPU-only setup since we only have a single-layer linear model.

---

[5]Our code is available at `https://anonymous.4open.science/r/mlrc-2021-exacerbating/`

# 5 Results

## 5.1 Results reproducing the original paper

In this Section, we are reporting the results for the two experiments conducted in the original paper.

### 5.1.1 Impact of the proposed attacks on fairness

First, we evaluate the effectiveness of the proposed adversarial attacks on the three datasets mentioned in Section 4.2 using the metrics discussed in Section 4.3, for varying $\epsilon$ values. We perform the anchoring attack, using both random (RAA) and non-random sampling (NRAA). We additionally reproduce Koh's influence attack [9] and Solan's attack [13], using our implementation. Our results are presented in Figure 1 and correspond to Figure 2 of the original paper.

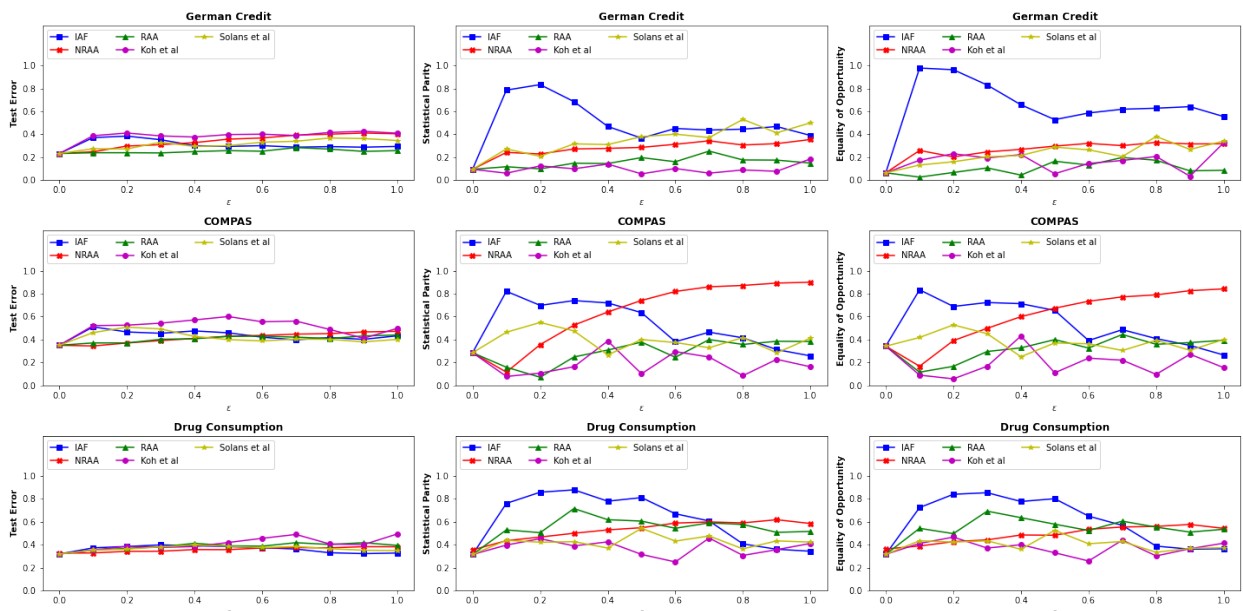

Figure 1: Impact on performance and fairness of a logistic regression classifier, using the attacks proposed in [11] and other state-of-the-art methods, for increasing $\epsilon$ values.

We observe that the IAF is the most versatile attack on fairness, as it can raise the test error by $20\%$ and push the SPD and EOD values close to $1$. This general trend matches the results of the original paper, although it appears that the effectiveness of the attack diminishes for higher values of $\epsilon$. As a result, we see cases where the fairness is impacted less than other attacks, which is contradictory to the results of the original paper.

The NRAA appears to be the second most effective fairness attack, especially for the COMPAS dataset, where it can reach the performance of the IAF, at the cost of using a significantly higher percentage of poisoned data $\epsilon$. However, it also appears to increase the model's test error up to $20\%$, which contradicts the findings of the original paper, that the NRAA attack does not affect performance.

Finally, the RAA appears to be less effective when compared to the NRAA. The test error was preserved, as in the original paper, but its impact on fairness was inconsistent depending on the value of $\epsilon$ and the dataset. It is worth mentioning that this attack exhibited the most variance in our results when using different seeds, which can be explained by the method's inherent stochasticity.

### 5.1.2 Regulation of the trade-off between impacting performance and fairness

We evaluated the regulation of the trade-off between impacting fairness and performance using the IAF on the same datasets and metrics as previously. Our results are presented in Figure 2 and, apart from our extra experiment for $\lambda = 0.5$, correspond to Figure 3 of the original paper.

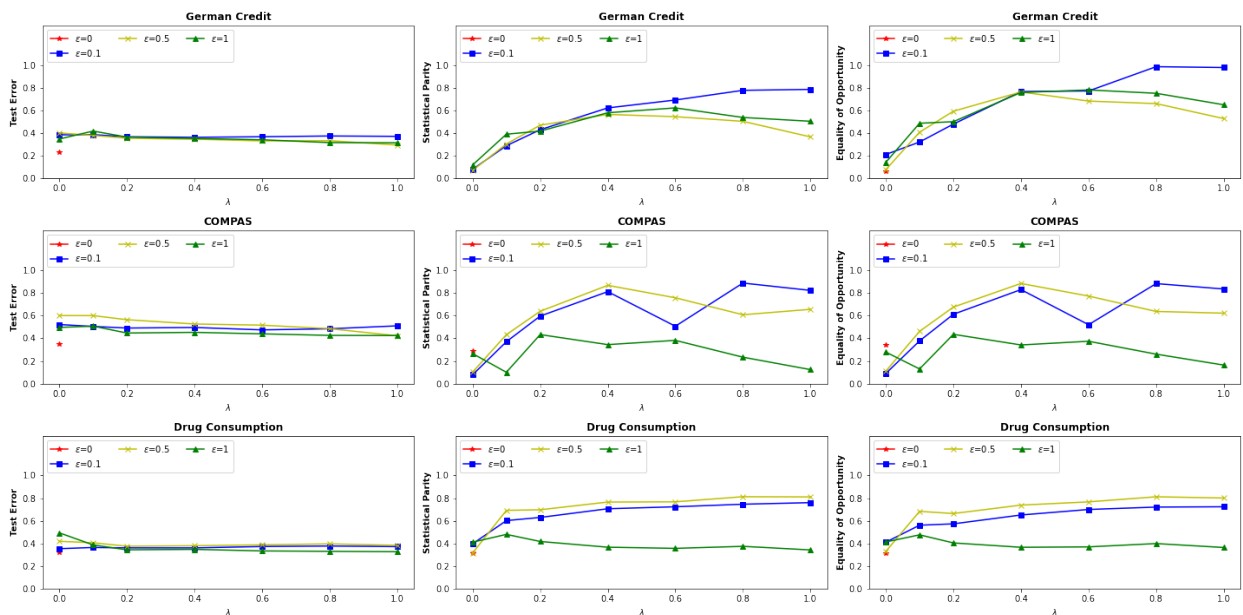

Figure 2: Impact on performance and fairness of a logistic regression model using the IAF, for increasing $\lambda$ values.

We notice that the IAF drops the model's performance by $10\%$ to $20\%$. The hyperparameters $\lambda$ and $\epsilon$ seem to not have a strong correlation with the test error, as every pair of them leave it intact. This comes in contrast to the results of the original paper, where higher $\lambda$ and $\epsilon$ values affect the performance less. We also observe that higher $\lambda$ values have a greater impact on fairness, which is in accordance with the original paper's results. However, in the original paper, higher $\epsilon$ values also increase the rate at which $\lambda$ affects the fairness of the targeted model, while in our results very high values, such as $\epsilon = 1$, seem to have the opposite effect.

## 5.2 Results beyond the original paper

In this section, we report our results for an additional experiment we conducted. Although there were many interesting directions we wanted to investigate, we focused on just one due to limited time and resources.

### 5.2.1 Inversion of the dataset's bias direction

The experiments of Section 5.1.1 made us question whether it is possible to use the principle idea behind the IAF to inverse the bias present in the datasets, instead of always exacerbating it in favor of the advantaged group. To this end, we changed the sign of $\lambda$, according to the intrinsic bias of the dataset. Our results are presented in Figure 3 and indicate this approach does indeed shift the bias of the dataset towards the other extreme.

An important byproduct of this technique is that it can be used to mitigate the existing bias of the datasets. We observe in Figure 3 that for $\lambda = 0.2$, the fairness metrics approach zero while the performance remains on the same level. Hence, tuning the value of $\lambda$ in a held-out validation set would allow us to augment the existing datasets to be fairer without sacrificing performance. Do note that in this experiment we used the actual differences of the SPD and EOD to better capture the direction of the bias. For more details, refer to Appendix C.

## 6 Discussion

Based on the results of the first experiment, we are able to partially verify the first two claims of the paper. More specifically, both the IAF and the NRAA are indeed the most effective attacks on fairness, under most of the evaluated settings. However, the RAA performs poorly compared to the existing methods, such as Koh's and Solan's, which contradicts part of the first claim. What is more, although the RAA does not affect the performance of the targeted system, the NRAA can, which contradicts part of the paper's second claim. Similarly, the results of our second

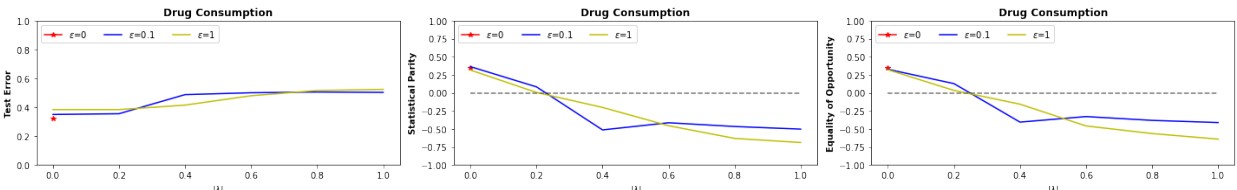

Figure 3: Reversing the intrinsic bias of the Drug Consumption dataset using a modified version of the IAF.

experiment suggest that although $\lambda$ is able to control the impact on fairness, it is not as effective in doing so with performance. Based on this, we can partially verify the third claim of the paper. In summary, although we were not able to fully verify the original claims based on our results, we can confirm the methods' effectiveness in attacking fairness.

## 6.1 What was easy

One of the things we found welcoming was the overall presentation of the paper which is nicely structured and has cohesive sections. The provided pseudo-code condenses the principal ideas of both attacks very intuitively, and the datasets used in the paper are publicly available and small in size. The latter welcomes everyone to reproduce the results, regardless of their computational budget. Additionally, the authors provide their code on GitHub where missing details can be found easily. All these elements hint at an easy reproduction of the results.

## 6.2 What was difficult

As we got familiarized with the concepts behind the attacks, we identified some issues which were not apparent at first. To begin with, even though the principle ideas are intuitive, the notation used is not always self-sufficient. The algorithms depend on other utilities (such as the projection of data in the feasible set) and non-trivial calculus operations, which are not discussed. Additionally, information about the model, training pipeline, hyperparameters, and data pre-processing used is absent. For these elements, we tried consulting the code provided by the authors, but it turned out to be convoluted. We encountered a structure that was hard to follow, non-intuitive variable names, absence of comments and docstrings, and large portions of unused code. All these elements made the reproduction of the results challenging and required some assumptions and critical decisions on our part.

## 6.3 Communication with the authors

We contacted the first author with a list of questions to resolve the existing ambiguities. Although the response was fairly quick, we were prompted to check the existing code in-depth, while further communication was discouraged.

# 7 Conclusion

In this reproduction study, we extensively reviewed the paper *Exacerbating Algorithmic Bias through Fairness Attacks*. We provided a clear foundation, upon which we described the proposed data poisoning attacks, namely the influence attack on fairness and the anchoring attack, as well as the experimental setup of the original paper. We filled in numerous details that we considered crucial for the reproducibility of the results. We evaluated the effectiveness of the proposed attacks both in terms of performance and fairness, and even though we did not manage to get the exact results of the original paper, our experiments show similar trends. Hence, we can verify the superiority of the proposed methods compared to the rival ones. Finally, we examined the regulation of the trade-off between impacting fairness and performance and found that while the impact on performance cannot be directly controlled, the impact in fairness can be. These findings suggest that although the original paper is not reproducible, its claims are valid.

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

## A List of inconsistencies, assumptions and corrections

After studying the original paper and the provided code, we spotted a few inconsistencies between the two. In order to deal with them, we had to make some assumptions that better aligned with the methods presented in the original paper.

Regarding the influence attack on fairness, Koh et al. [9] suggest that the train set during the attack is not $\mathcal{D}_c \cup \mathcal{D}_p$, but $B(\mathcal{D}_c \cup \mathcal{D}_p)$, i.e. the set that passes from the defense mechanism $B$. Hence, we assume that when the authors mention that they *update the feasible set* $\mathcal{F}_\beta \leftarrow B(\mathcal{D}_c \cup \mathcal{D}_p)$, they mean that they update the parameters $\beta$ of the feasible set. Additionally, pre-computing $H_{\hat{\theta}}^{-1}$ is computationally expensive and is avoided in the authors' code. Instead the computational trick introduced in Koh et al. [8] is used.

Regarding the anchoring attack, we noticed two issues in the paper and the accompanied code. The anchoring attack with non-random sampling is deterministic and thus each iteration of attack will result in the same poisoned dataset $\mathcal{D}_p$ discarding the need to have multiple iterations. Moreover, the anchoring attack with random sampling is a stochastic method, yet in the existing implementation, the random number generator is seeded with the same number in every iteration, resulting in the same poisoned dataset $\mathcal{D}_p$. As a result, the attack's output will be deterministically generated as the method's stochasticity is discarded with the iterations being redundant.

Regarding the helper functions for both attacks and defenses, it seems that the authors use the LP relaxation technique implemented in [9] by default in their experiments. However, we could not find an explicit mention of this in paper. Additionally, we did not find any suggestion for choosing the neighbor cutoff radius $\sigma$, which seems to be hard-coded for every dataset. Finally, the choice of radii for the L2 constraint and slab cutoff are not discussed in the paper, although the authors seem to use similar techniques to the ones discussed in 3.2.

Regarding the pre-processing pipeline applied to the original data, we noticed it is neither mentioned in the paper nor provided in the GitHub repository of the authors. After contacting them, they pointed us to another repository that included a similar pre-processing pipeline to the one applied for the paper. Observing the code, we noticed two issues. Categorical data were converted to one-hot encoded and then standardized with the quantitative features, which is not the most efficient technique. Also, the test data were normalized along with the train data, allowing information from the test set to be utilized for training.

Regarding the experimental setup, the reported results in the paper are the output of a single seed for the random generator. As a consequence, there was only a single split of the data between training and testing leading to results with high variance.

## B Finding the most popular point in a dataset

Let $\mathbf{x}_1, \mathbf{x}_2, \ldots, \mathbf{x}_n \in \mathbb{R}^m$ be points in a dataset $\mathcal{X}$. Our goal is to define the most popular point $\mathbf{x}_{\text{pop}}$ in a meaningful way, such that it is dataset agnostic, i.e. it does not require manual input of parameters, such as a manually defined radius for each dataset. We mainly experimented with two methods.

- **Method A: Percentile Radius**: We define the most popular point

$$\mathbf{x}_{\text{popA}} \overset{\text{def}}{=} \underset{\mathbf{x} \in \mathcal{X}}{\arg\max} \, \text{CountN} \, (\mathbf{x}, R) \tag{4}$$

  where $\text{CountN}(\mathbf{x}, R)$ is a function that returns the number of points $\mathbf{x}_i \in \mathcal{X}$ such that $d(\mathbf{x}, \mathbf{x}_i) \le R, \; R \in \mathbb{R}^+ \; \forall \mathbf{x}_i \in \mathcal{X}$ for some distance metric $d$. The problem of picking a fitting radius $R$ is not trivial as the radius has to be neither too small nor too big as either all or no points would be considered neighbors, respectively. The method we propose is to pick a radius $R$ such that at least $\alpha\%$ of $\mathbf{x} \in \mathcal{X}$ satisfy $||\mathbf{x} - \boldsymbol{\mu}|| \le R$, where $\boldsymbol{\mu}$ the centroid of $\mathcal{X}$. In our experiments, $\alpha = 15$ has proved to be decent for all three datasets.

- **Method B: Exponentially decayed distances**: We define the most popular point

$$\mathbf{x}_{\text{popB}} \overset{\text{def}}{=} \underset{\mathbf{x} \in \mathcal{X}}{\arg\max} \sum_{\mathbf{x}' \in \mathcal{X}} \exp \left( -\frac{d(\mathbf{x}, \mathbf{x}')}{\sigma_{d(\mathcal{X})}^2} \right) \tag{5}$$

  where $d$ is a distance metric and $\sigma_{d(\mathcal{X})}^2$ denotes the variance of all the distances of the points in the dataset to each other under $d$. We define $\sigma_{d(\mathcal{X})}^2 \overset{\text{def}}{=} \text{Var} \, (\text{vec}(d(\mathcal{X})))$, where $[d(\mathcal{X})]_{ij} := d \left( [\mathcal{X}]_{i:} , [\mathcal{X}]_{j:} \right)$.

In Method A, we still define neighbors based on balls surrounding datapoints. Even though we still have to pick an $\alpha$, the choice is easier, as we don't have to manually check the distances in the dataset.

In Method B, we discard the idea of neighbors based on radii around points and we turn our focus on finding a datapoint in a very dense area of the dataset. To ensure that the sum is higher for points with a lot of other points in their close vicinity, we exponentially decay the distances. This forces points close to our point in question to contribute more to the sum. We also need the method to be dataset agnostic, thus we need to scale the wideness of the exponential kernel. If the variance[6] of the distances is high, we need to widen the kernel such that points further away still contribute to the sum. In contrast, if distances have low variance we need to sharpen the exponential kernel to make sure that only points close enough to the point in question contribute to the sum. We define the variance of the dataset $\mathcal{X}$ as $\sigma^2_{d(\mathcal{X})} = \mathrm{Var}\left(\mathrm{vec}(d(\mathcal{X}))\right)$, where $[d(\mathcal{X})]_{ij} := d\left([\mathcal{X}]_{i:}, [\mathcal{X}]_{j:}\right)$. We opted for this method since it requires the least amount of arbitrary assumptions about the dataset. Preliminary experiments hinted towards method B achieving slightly better results in our task, but this wasn't pursued further.

In the Anchoring Attack, we need to sample a negative sample $x_{\mathrm{target}-}$ from the advantaged class $\mathcal{D}_{\mathrm{adv}}$ and a positive sample $x_{\mathrm{target}+}$ from the disadvantaged class $\mathcal{D}_{\mathrm{disadv}}$. In the non-random sampling setting (NRAA), we simply calculate the most popular point in the negative but advantaged class $\mathcal{D}_{\mathrm{adv}} \cap \mathcal{D}^- \subset \mathcal{D}$ and the most popular point in the positive but disadvantaged class $\mathcal{D}_{\mathrm{disadv}} \cap \mathcal{D}^+ \subset \mathcal{D}$.

# C  Data Augmentation

As it has been demonstrated through experimental evaluation, the IAF can deteriorate a model's fairness. However, we argue that the same approach can be applied for data augmentation to increase a model's fairness resulting in an unbiased classifier.

The use of the fairness metrics with absolute values, as described in Section 4.3, fails to highlight the bias direction. However, by using the actual differences of the metrics, we can utilize this information. Therefore, knowing the initial bias of the data by inspecting the sign of $P(y_{\mathrm{label}} \mid \mathbf{x} \in \mathcal{D}_{\mathrm{adv}}) - P(y_{\mathrm{label}} \mid \mathbf{x} \in \mathcal{D}_{\mathrm{disadv}})$, we can assume that the model's bias will be in the same direction, i.e., the SPD and EOD will have the same sign. To this end, to direct a model's bias towards zero, we have to use the opposite sign of the aforementioned quantity for the values of $\lambda$.

Moreover, as the altered method is used for augmentation, the test dataset $\mathcal{D}_{\mathrm{test}}$ should not be utilized, in contrast with the IAF. Finally, we could use a validation set to halt the data augmentation process in order to find the optimal value of $\lambda$ where the SPD and EOD would be close to zero.

---

[6] The mean of the dataset or some other statistic could also be used, which intuitively makes more sense. Basic experiments hinted that dividing by the variance performed better, but the mean method can not be completely discarded as we didn't conduct thorough experiments due to time constraints.

