# OpenReview forum: "[Re] Exacerbating Algorithmic Bias through Fairness Attacks"
_ML_Reproducibility_Challenge/2021/Fall — RC2021_

### Official Review · Reviewer_Y6FW · 2022-02-28
**Review of [Re] Exacerbating Algorithmic Bias through Fairness Attacks**

**Rating:** 9
**Confidence:** 4

**Review:**

This paper focused on reproducing the results from "Exacerbating Algorithmic Bias through Fairness Attacks" (Mehrabi, 2020). Overall, the paper was very easy to read and had good organization, and expanded upon the original findings to their own explorations.

Pros
+ The paper did a good job of covering reproducibility summary, scope, communication with original authors, discussion on results, recommendations, and results beyond the paper.
+ The authors went beyond the findings of the original paper and extended the evaluation set up to test whether current methods can be used to invert the inherent bias of a dataset.
+ Provides an easy way for future researchers to more easily reproduce the paper as well as extend it with the newly added code.

Cons
- Explanations for why the original paper results deviated from the reproduced paper results would have been helpful, especially since for some tests, the results seemed to have the opposite effect (value of e for example)

Questions
- Are the hyperparameters used for reproducing the paper the same ones used in the original paper?

---

### Official Review · Reviewer_Vb1a · 2022-03-07
**Nice reproduciblity report with convincing result and clear writing.**

**Rating:** 7
**Confidence:** 3

**Review:**

- The scope of reproducibility is clearly presented.

- It seems the author does not reuse the original paper's code and re-implement the experiments via Pytorch-lightning. The author's codebase is provided with clear documentation.

- The author makes a reasonable effort for communicating with original authors.

- The report provides a study beyond the original paper which investigates the effectiveness of the original technique for inverting the dataset’s bias direction.

- It seems the author decide their own hyper-parameters in their experiments. As reported, the numbers are not the same as the original paper. But original paper's claims are verified.

- The submission is well written and organized.

---

### Meta-Review · Area_Chair_kwih · 2022-04-08

**Recommendation:** Accept
**Confidence:** 4

**Metareview:**

A very strong reproducibility study that verifies most of the claims made in the original paper. The authors also present results beyond the paper to investigate the effectiveness of the method which is impressive. It seems like the authors use a different set of hyperparameters as opposed to the original paper and we feel there should be an explanation highlighting the reason for this.

---

### Decision · Program_Chairs · 2022-04-09

**Decision:**

Accept

**Comment:**

Following the recommendation of reviewers and meta-reviewer, the paper is accepted for ML Reproducibility Challenge 2021, and will be published in the upcoming special edition of ReScience Journal.